# Degradation of benzo[a]pyrene by halophilic bacterial strain *Staphylococcus haemoliticus* strain 10SBZ1A

**Alexis Nzila**[1]*, **Musa M. Musa**[2], **Saravanan Sankara**[1], **Marwan Al-Momani**[3], **Lei Xiang**[4], **Qing X. Li**[5]

**1** Department of Life Sciences, King Fahd University of Petroleum and Minerals, Dhahran, Saudi Arabia,
**2** Department of Chemistry, King Fahd University of Petroleum and Minerals, Dhahran, Saudi Arabia,
**3** Department of Mathematics, King Fahd University of Petroleum and Minerals, Dhahran, Saudi Arabia,
**4** Guangdong Provincial Research Center for Environment Pollution Control and Remediation Materials, College of Life Science and Technology, Jinan University, Guangzhou, China, **5** Department of Molecular Biosciences and Bioengineering, University of Hawaii at Manoa, Honolulu, Hawaii, United States of America

* alexisnzila@kfupm.edu.sa

**Data Availability Statement:** All relevant data are within the paper and its Supporting Information files.

## Abstract

The exploitation of petroleum oil generates a considerable amount of "produced water or petroleum waste effluent (PWE)" that is contaminated with polycyclic aromatic hydrocarbons (PAHs), including Benzo[a]pyrene (BaP). PWE is characterised by its high salinity, which can be as high as 30% NaCl, thus the exploitation of biodegradation to remove PAHs necessitates the use of active halophilic microbes. The strain 10SBZ1A was isolated from oil contaminated soils, by enrichment experiment in medium containing 10% NaCl (w/v). Homology analyses of 16S rRNA sequences identified 10SBZ1A as a *Staphylococcus haemoliticus* species, based on 99.99% homology (*NCBI, accession number GI*: MN388897). The strain could grow in the presence of 4–200 µmol l⁻¹ of BaP as the sole source of carbon, with a doubling time of 17–42 h. This strain optimum conditions for growth were 37 °C, 10% NaCl (w/v) and pH 7, and under these conditions, it degraded BaP at a rate of 0.8 µmol l⁻¹ per day. The strain 10SBZ1A actively degraded PAHs of lower molecular weights than that of BaP, including pyrene, phenanthrene, anthracene. This strain was also capable of removing 80% of BaP in the context of soil spiked with BaP (10 µmol l⁻¹ in 100 g of soil) within 30 days. Finally, a metabolic pathway of BaP was proposed, based on the identified metabolites using liquid chromatography-high resolution tandem mass spectrometry. To the best of our knowledge, this is the first report of a halophilic BaP degrading bacterial strain at salinity > 5% NaCl.

## Introduction

The exploitation of oil is associated with the generation of wastewater, also called produced water or petroleum waste effluent (PWE) or reservoir water, at a ratio of three barrels of PWE for one barrel of exploited oil. Based on an estimate of 90 million barrels of oil extracted daily, a staggering volume of more than 250 million barrels of PWE are daily generated worldwide

**Funding:** Alexis Nzila (King Fahd University of Petroleum and Minerals, KFUPM) would like to acknowledge the support of the Deanship of Scientific Research (DSR) at KFUPM, under grant IN171022. Qing Li (University of Hawaii, USA) acknowledges the support of the US Department of Agriculture, under USDA grant HAW5032-R, for high resolution liquid chromatography mass spectrometry analysis.

**Competing interests:** NO authors have competing interests

[1]. PWE is hypersaline, up to 30% (w/v) NaCl, and is heavily contaminated with petroleum products [1, 2]. Other hypersaline environments contaminated with petroleum products can be industrial effluents and salt marshes [3–5].

These pollutants can be removed by bioremediation, a process that employs microorganisms to degrade toxic pollutants to harmless products, and when this process is complete, $CO_2$ is generated as the end product. This strategy is environmentally safe and cost-effective. Thus, the exploitation of bioremediation in removing pollutants from PWE and salt marshes requires the use of halophiles, microbes that can actively thrive in media containing high salt concentrations.

Polycyclic aromatic hydrocarbons (PAHs) are classified as low molecular weight PAHs, containing three or less fused benzene rings such as naphthalene, anthracene and phenanthrene, and high molecular weight PAHs (HMW-PAHs), consisting of four or more fused benzene rings, such as pyrene and benzo[a]pyrene (BaP) [6]. The biodegradation of HMW-PAHs is challenging, the higher the number of rings, the more difficult it is to degrade [7]. Because of its high ring number, BaP is one of the most recalcitrant PAHs to degradation, thus, it persists longer in the environment, with its attendant toxicity. BaP is ranked as number 8 out of 275 chemicals on the priority list of hazardous environmental substances [8]. This compound is toxic to both marine flora and human, moreover, it is carcinogenic and can lead to developmental, neurological, reproductive and immunological toxicities [9, 10]. Thus, removal of this pollutant from the environment remains a priority.

Several studies have been dedicated on the degradation of PAHs by active non-halophilic microorganisms [11–13]. Although most of this work has been devoted to PAHs containing up to four fused benzene rings, nevertheless, few microbes have been reported to degrade BaP in non-halophilic conditions, they include *Sphingomonas yanoikuyae* JA, *Mycobacterium* sp., *Mycobacterium vanbaalenii*, *Stenotrophomonas maltophilia*, *Novosphingobium pentaromativorans*, *Mesoflavibacter zeaxanthinifaciens*, *Ochrobactrum* sp., *Bacillus licheniformis* and *Bacillus subtilis* [14–22].

The degradation of PAHs containing up to four fused benzene rings has also been reported in salinity conditions by various halophilic microbes [3, 23, 24]. However, reports are scanty on BaP degradation by halophilic microbes. Aruzahgan et al. reported a stain of *Ochrobactrum* sp. VP1 capable of using BaP as the sole source of carbon, in moderate salinity conditions of NaCl (3%, w/v) [25]. A consortium of bacteria (*Achromobacter* sp. AYS3, *Marinobacter* sp. AYS4 and *Rhodanobacter* sp. AYS5) has also been reported to degrade BaP in the presence of phenanthrene using the same salinity conditions (NaCl, 3%, w/v) [26]. To the best of our knowledge, so far, only one BaP-degrading strain, *Ochrobactrum* sp.VP1, has been reported to utilise BaP as a single strain, but in relatively low salinity of 3% (w/v).

As part of the current work, a halophilic bacterium capable of degrading BaP in the presence of 10% (w/v) NaCl has been isolated and characterised. The potential of this strain in removing BaP in contaminated soil samples has also been evaluated. BaP metabolites were investigated using high-performance liquid chromatography (HPLC), coupled with high-resolution tandem mass spectrometry.

## Materials and methods

### Chemicals

Chemicals for the Luria-Bertani Broth (LB) culture medium were obtained from Difco (Lawrence, Kansas, USA). $(NH_4)_2SO_4$, $KH_2PO_4$, $CaCl_2 \cdot 7H_2O$, $MgSO_4 \cdot 7H_2O$, $Na_2HPO_4$ and $FeSO_4 \cdot 7H_2O$, BaP, pyrene, anthracene, phenanthrene, naphthalene, phthalic acid, salicylic acid, catechol; palmitic, oleic and stearic acids, were purchased from Sigma-Aldrich (St. Louis, MO,

USA). Luria-Bertani Broth (LB) was purchased from Difco (Lawrence, Kansas, USA). All the chemicals were of analytical grades.

## Microbial isolation

The isolation of bacteria that degrade BaP was carried out using soil contaminated from a filling petrol station, located at the King Fahd University of Petroleum & Minerals, Dhahra (Saudi Arabia). The enrichment culture was carried out in 50 ml Bushnell-Hass medium containing BaP (BH-BaP), which consists of $(NH_4)_2SO_4$ (2.38 g), $KH_2PO_4$ (1.36 g), $CaCl_2 \cdot 7H_2O$ (10.69 g), $MgSO_4 \cdot 7H_2O$ (0.25 g), $Na_2HPO_4$ (1.42 g), $FeSO_4 \cdot 7H_2O$ (0.28 mg) per liter, and supplemented with 0.05% (w/v) [2 mmol $l^{-1}$] of BaP as a sole carbon source and 1 g of contaminated soil. This Enrichment was carried out in 3 different salinity conditions, by adding 10, 15 and 20% NaCl (w/v) in the culture. The cultures were carried out at 37 °C, at 120 rpm for 3–4 weeks, and then transferred in a fresh culture medium (1/10, v/v) for 2–3 weeks. This step was replicated 4–5 times, until the growth of bacteria was observed. Bacterial colonies were separated using solid agar culture, prepared in BH-BaP medium (1%, w/v), and then incubated at 37 °C for 15–21 days. The purity of the isolated individual colonies was confirmed by another solid agar culture (in the same conditions), and thereafter, these colonies were cryopreserved in 15% (v/v) glycerol.

## Bacterial count

Bacterial count was carried out in solid agar culture plate, prepared in rich LB medium (1%, w/v), containing NaCl at the optimum concentration of each tested strains. A series dilution of bacterial culture was prepared by a dilution of factor of 10, 100 and 1000, in BH medium. Therefore, one ml of each bacterial culture was spread separately in agar solid plates, and incubated at 37 °C. After 24H, plates that had clear colonies were counted, and the results were represented as colony forming units per ml (CFU $ml^{-1}$).

## Scanning electron microscopy (SEM) analysis

Bacteria were observed under the electron microscope JSM-T300 (JEOL, Japan), following by their fixation in formaldehyde (5%, v/v) for 12 h, a serial dehydration in 30%, 50%, 70%, 80%, 90% and 95% of ethanol, (v/v), and gold coating, according to a protocol described elsewhere [27].

## Species identification

Species identification was carried out using 100ml of bacterial culture grown in LB rich medium. After 24H, the culture was centrifuged at 5000 g for 5 min at 4˚C, and the corresponding pellet, which consisted of bacteria, was isolated. Thereafter, bacteria DNA was extracted and purified using Qiagen Powerfecal Kit (Hilden, Germany), following the user's manual. The resulting purified DNA was subjected 16S rRNA gene amplification by PCR and sequencing, as described previously [28]. Around 1400 base pairs of 16S rRNA gene was amplified using Primers 27F AGAGTTTGATCMTGGCTCAG and 1492R CGGTTACCTTGT TACGACTT, and then sequenced using the primers 518F: CCAGCAGCCGCGGTAATACG and 518R: GTATTACCGCGGCTGCTGG, a Big Dye terminator cycle sequencing kit (Applied Bio-Systems, USA), and an automated DNA sequencer (Applied BioSystems model 3730XL, USA). The use of Basic Local Alignment Search Tool (BLAST), available at the National Center for Biotechnology Information (NCBI) database, permitted species identification.

## Bacterial growth in the presence of PAHs and various hydrocarbon molecules, and effect of pH and salinity

To carry out these experiments, around $5 \times 10^5$ colony forming units (CFUs) of bacteria, which were pre-cultured in an LB medium, were then cultured in 50 ml BH medium containing BaP or other substrates. The polar substrates salicylic acid and catechol, and the aliphatic compounds were directly added to the medium, while non-polar substrates (BaP, pyrene, phenanthrene, naphthalene and anthracene) were dissolved first in dimethyl sulfoxide (DMSO), prior to their addition in BH culture medium. Bacterial growth in the presence of BaP was also assessed at 30, 35, 40 and 45 $^{\circ}$C; salinity of 0, 5, 10, 15 and 20% of NaCl (w/v), and pH 5, 6, 7, 8 and 9. CFU ml$^{-1}$, which reflects bacterial counts, was employed as a measure of growth.

These counts were then fitted in the growth curves $Q_t = Q_o e^{-kt}$ equation, so as to compute the doubling time (dt). $Q_t$ and $Q_o$ represent the bacterial count at time t, and time 0 respectively, and $k$ is the growth rate. The growth rate ($k$) was then used to compute dt values (in hours) according to the equation: dt = ln(2)/$k$. All experiments were carried out in duplicate.

## Quantification of BaP

The quantification of the remaining BaP was carried over a period of 20 days, in a 100 ml containing 20 µmol l$^{-1}$ BaP, as reported previously (Budiyanto et al. 2017). Around 100 ml culture samples were collected every 5 days, and the remaining BaP extracted using ethyl acetate (50 ml × 2), after sonication for 30 min. After dehydration with calcium chloride, the resulting organic layers were dried under vacuum, and then dissolved in 500 µl of chloroform, followed by gas chromatography mass spectrometry (GC-MS) analysis (GC, Agilent 6890N, MS, Agilent 5975B). A 5-point standard curve of BaP (4, 20, 100, 200, 400 µmol l$^{-1}$) were used to assess the concentration of the remaining BaP, which were then fitted in the exponential decay equation $Q_t = Q_o e^{-kt}$ (Lu et al. 2014), for the computation of ($k$), representing the biodegradation rate.

## Quantification of BaP utilisation in soil samples

Around 100 g of soil sample was spiked with 20 µmol of BaP, and placed in a 5 cm x 5 cm Petri dish. This soil was kept wet following an addition of around 5 ml of BH medium, at pH 7 and salinity 5%, and an approximate amount of $10^7$ CFU was added, and then incubated at 37 $^{\circ}$C. Control experiments were prepared in the same conditions, but without adding bacteria. At day 0, 15 and 30, each sample (from 5 cm x 5 cm Petri dish) was sonicated and extracted as explained in the previous section, and then subjected to GC (Agilent 7890A) equipped with a flame ionization detector (FID) for quantification.

## Identification of metabolites by HPLC-MS/MS

Metabolites were identified following a 15-day bacterial culture in the presence of 4 mmol l$^{-1}$ BaP. The medium was then extracted with ethyl acetate (300 ml × 2), and the organic layers was concentrated to around 200 ml, and extracted with 1.0 M sodium hydroxide (200 ml × 2). The resulting aqueous layer was neutralised with concentrated hydrochloric acid, then extracted with ethyl acetate (150 ml × 2). The organic layer was dried with sodium sulfate, and then evaporated under vacuum. The sample was dissolved in 2 ml of 30% (v/v) methanol in water before analysis using a Shimadzu Nexera Prominence LC, interfaced with an AB SCIEX X500 QTOF mass spectrometer. LC separation was conducted using an Agilent HC-C$_{18}$ column (4.6×250 mm, Agilent, USA). Methanol and water were set as the mobile phase A and B, respectively. In the gradient program, the potential metabolites were eluted by a linear gradient of mobile phase B, with a flow rate of 1.0 ml/min. The mobile phase B started at 50%, increase

to 95% at 60 min, and then back to 50% at 60.1 min (held for 4.9 min), with a total run time of 65 min. The ion source parameters were set as 300˚C, 30, 60 psi, and 60s for ion source temperature, curtain gas, ion source gas 1, and ion source gas 2, respectively. To improve the chances of observing potential metabolites, both negative and positive TOFMS/MS scan modes were applied. In the negative scan mode, the typical TOFMS/MS parameters were as follows: ion spray voltage (IS), -4500 V; CAD gas, 7; TOF start mass, 50 Da; TOF start mass, 1000 Da; accumulation time, 0.15 s; declustering potential, -60 V; declustering potential spread, 0 V; collision energy, -10 V; collision energy spread, 0 V. Except ion spray voltage (IS, 5500 V), declustering potential (60 V), and collision energy (10 V), the other typical TOFMS/ MS parameters in the positive scan mode were same with those in the negative scan mode.

## Statistical analyses

One-way analysis of variance (ANOVA), t-test and a linear regression fitting model were employed to analyse the data, using MINITAB (Version 16, Coventry, UK). Pearson correlation coefficient was used to establish the data strength of linearity, and the $p < 0.05$ was the level of significance in all tests.

## Results

### Enrichment, strain isolation and species identification

No bacterial growth was observed in the enrichment protocol with 15 and 20% NaCl salinity, in the presence of BaP as a sole carbon source. In the same condition, one single colony (10SBZ1A) grew at 10% NaCl. Under light microscope (40x-1000X), 10SBZ1A colony was white/cream, with circular form and flat with entire margins. These bacteria are Gram-positive, and electron microscopy shows that they have a coccus-shape, with about 1.2 μm diameter (**S1 Fig**).

Bacterial DNA was isolated and purified, and subjected to 16S rRNA gene sequencing for species identification. Using BLAST program for homology analysis of available 16S rRNA gene sequences in the NCBI database, this strain 10SBZ1A was identified as *Staphylococcus haemoliticus*, based on the threshold of 99% homology (NCBI accession number GI: MN388897.

### Effect of BaP concentration on bacterial growth and BaP degradation

The effect of various concentrations of BaP (4, 20, 40, 100, 200, 400 μmol l⁻¹) was assessed on the bacterial growth at pH 7, temperature 37 °C and salinity 10% NaCl. At BaP concentrations of 4–200 μmol l⁻¹, the strain showed a rapid growth, reaching the maximum growth within 6–8 days, with the maximum count ranging between $10^7$ and $10^8$ CFU ml⁻¹ (**Fig 1**). However, at 400 μmol.l⁻¹, the bacterial growth rate decreases, as shown by the time delay at which the maximum growth was achieved, which was at day 15. This BaP inhibitory effect was also confirmed by the increase in the culture doubling time (dt) as the BaP concentration increases (**Fig 2**). At 4–40 μmol l⁻¹, dt values were in the range of 17–24 h, and these values almost doubled at 100–400 μmol l⁻¹. The single regression ANOVA showed the correlation between dt values and BaP concentrations is statistically significant ($p < 0.05$), and it follows a linear equation dt = 20.83+0.0705×C ($R^2$ = 0.78, where C is BaP concentration). All subsequent experiments were carried out at 40 μmol l⁻¹ of BaP, unless otherwise stated.

### Effect pH on bacterial growth and BaP degradation

The ability of 10SBZ1A to degrade BaP was tested at pHs 5, 6, 7, 8, and 9, while the temperature was fixed at 37 °C, and salinity at 10% NaCl (**Table 1**). At pH 7, dt value was 21.49 ± 0.98,

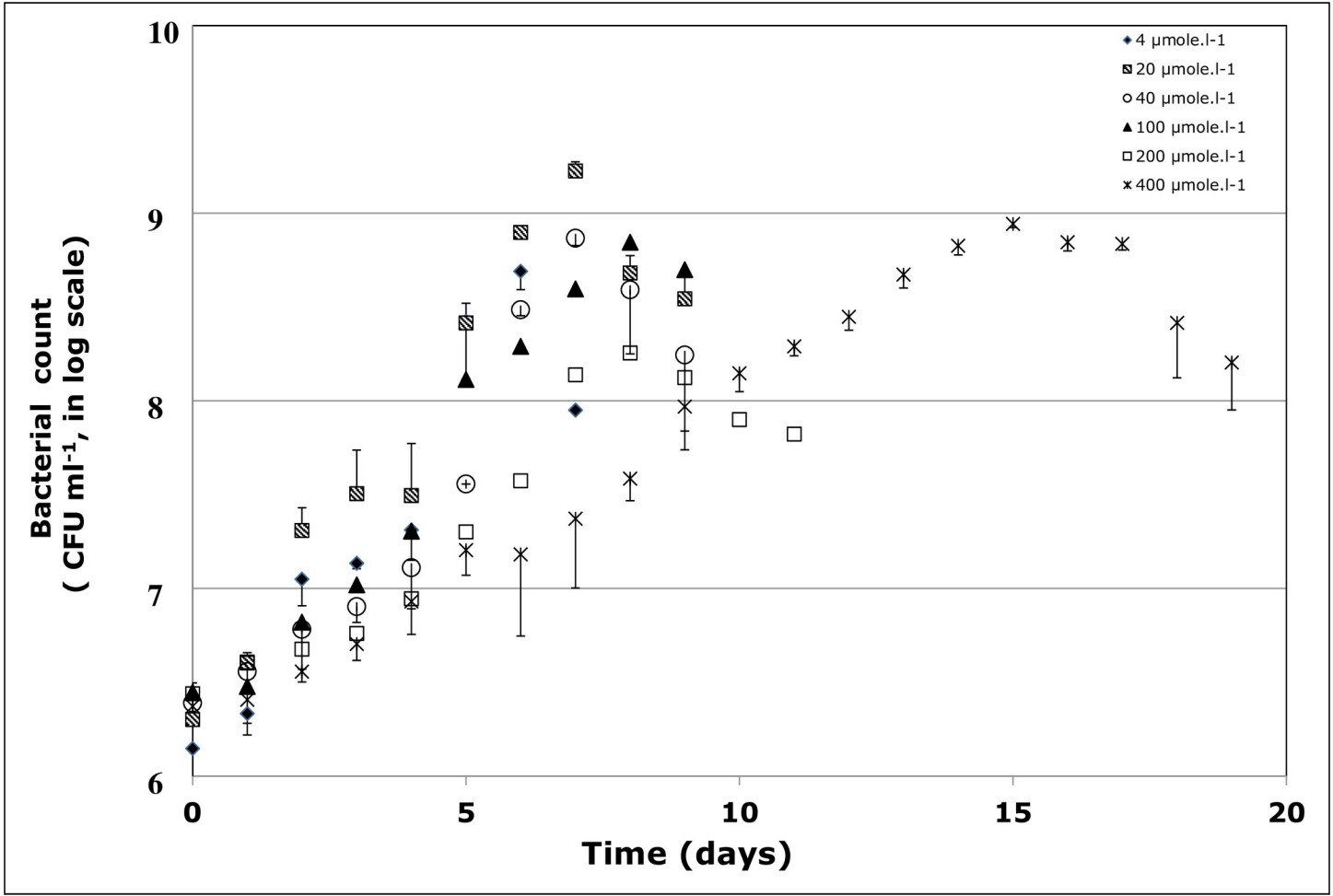

**Fig 1. Growth profile of *Staphylococcus haemoliticus* 1OSBZ1A strain in the presence of various concentrations of benzo(a)pyrene (BaP).** CFU represents colony forming unit.

however, at other pH values, the rates of growth were so slow that dt could not be computed. Thus, this strain could actively degrade BaP in neutral medium (i.e., pH 7).

## Effect of temperature on bacterial growth and BaP degradation

The effect of temperature on BaP degradation was evaluated at 35, 37, 40 and 45 °C (pH 7, salinity 10% NaCl) [Table 1]. Overall, dt fell in the range 21–29 h, and the temperature 37 °C corresponded to the smallest dt (21±1 h). Thus, 37 °C was considered the optimum for the degradation of BaP using the strain 10SBZ1A (Table 1), although these dt differences were statistically significant between 35 and 37 °C only (p<0.05).

## Effect of salinity on bacterial growth and BaP degradation

The strain 10SBZ1A was isolated from a medium containing 10% NaCl (wt/v). To establish its salinity tolerance, its growth was assessed at 0, 5, 15 and 20% NaCl (while keeping the temperature at 37 °C and pH 7), and the results were compared with that obtained at 10% NaCl. No growth was observed at 0 and 20% NaCl. The highest growth rate, as measured by dt values

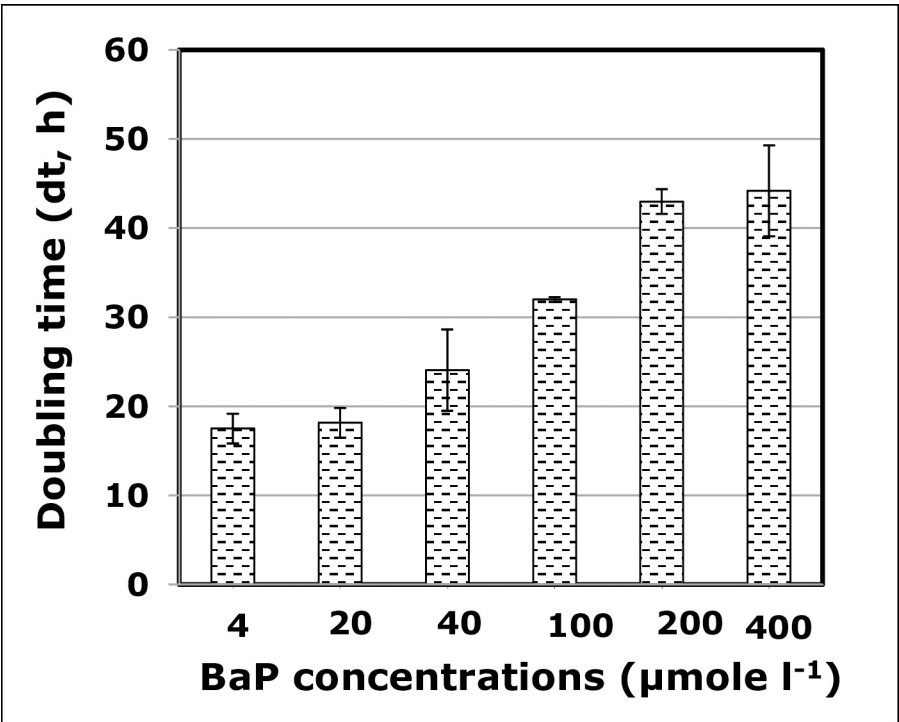

**Fig 2. Doubling time (dt, in hours) of culture of *Staphylococcus heamolysis* strain 10SBZ1A in the presence of various concentrations of Benzo(a)pyrene (BaP).**

(21±1 h), was observed at 10% NaCl, followed by a dt of 47±7 h at 5% NaCl and 156±74 h at 15% NaCl. Thus, these data show that the optimum salinity of *S. haemolyticus* is 10% NaCl, although these difference were not statistically significant (p>0.05).

**Table 1. Doubling time (dt, in hours) of the degradation of benzo(a)pyrene (BaP) by *Staphylococcus haemoliticus* strain 10SBZ1A, as a function of pH, temperature and salinity).**

|  | Conditions | Doubling time (dt, h) |
|---|---|---|
| pH | 5 | ND[a] |
|  | 6 | ND |
|  | 7 | 21.5 ± 1.0 |
|  | 8 | ND |
|  | 9 | ND |
| Temperature | 35 °C | 28.5 ± 2.4[b] |
|  | 37 °C | 21.5 ± 1.0 [b] |
|  | 40 °C | 24.5 ± 1.0 |
|  | 45 °C | ND |
| Salinity (% NaCl) | 0 | ND |
|  | 5% | 46.8 ± 7.4 |
|  | 10% | 21.5 ± 1.0 |
|  | 15% | 156 ± 74 |
|  | 20% | ND |

[a] Not determined due to slow growth rates.

[b] The difference of dt values were statistically significant ($p < 0.05$).

## Utilisation of others PAHs

The ability of the strain 10SBZ1A to degrade PAHs that are smaller than BaP, such as pyrene, phenanthrene, anthracene and naphthalene, was assessed, at pH 7, temperature 37 °C and salinity 10% NaCl. Monocyclic aromatic compounds salicylic acid and catechol were also included, along with the aliphatic and long chain palmitic, stearic, and oleic acids. All these substrates were assessed at 40 μmol l$^{-1}$. As shown in **Fig 3**, overall, the ability of 10SBZ1A to degrade PAHs increases as the PAH ring number decreases. The dt of this strain in the presence of BaP was around 42 h, and although it is slightly similar to that with pyrene (a four-ring-containing PAH), however, this value decreases to 25–30 h for phenanthrene, anthracene, and naphthalene. The lowest dt value was associated with catechol (around 24 h) (**Fig 3**). A Significant linear correlation between dt values and the substrate's number of aromatic rings was observed, and this correlation follows the linear equation: dt = 18.77+4.636×N (p <0.001, $R^2$ = 66%) [N represents the number of rings].

The data also showed the higher efficiency of this strain to degrade long chain aliphatic acids compared to aromatic compounds. Indeed the dt for palmitic, oleic and stearic acids were 19±0.7, 17±0.9 and 15±0.8 h, respectively.

## Degradation rate

The degradation profile shows that this strain degrades 50% of 20 μmol.l$^{-1}$ BaP at day 12.5, and at day 25, almost 80% of BaP was degraded, leading to a BaP degradation rate of 0.8 μmol l$^{-1}$. day$^{-1}$ (**Fig 4**). Around 20% of abiotic degradation was observed (as indicated by the control).

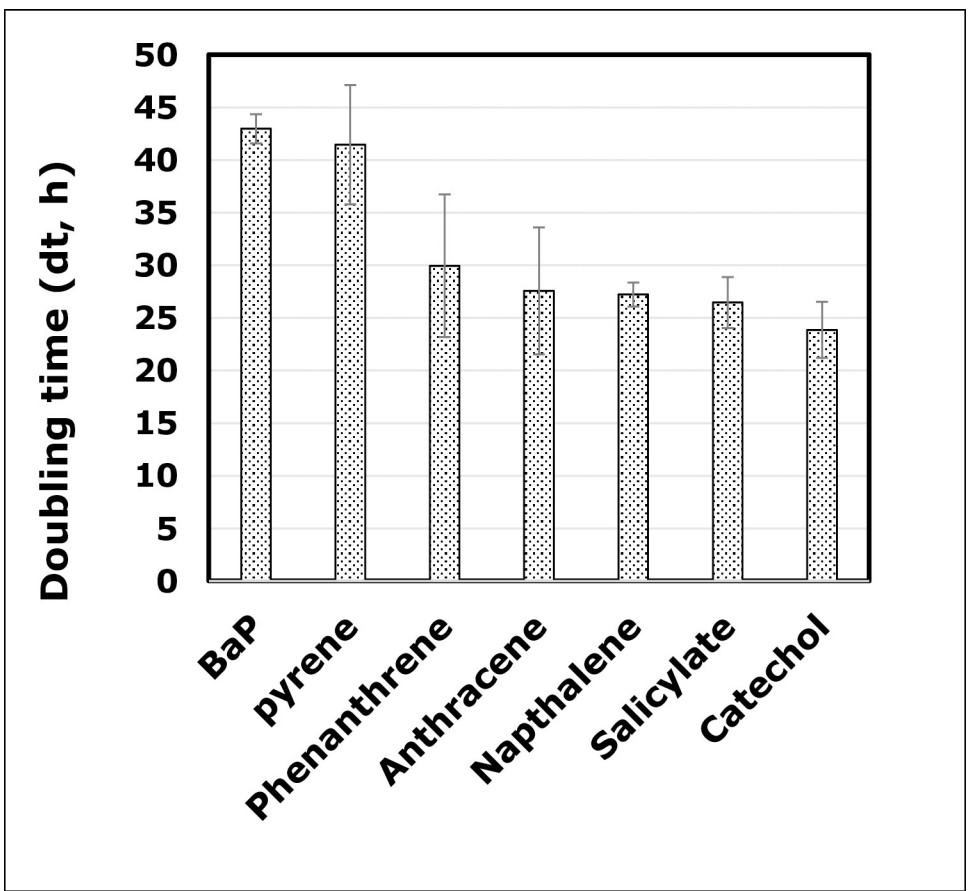

**Fig 3. Doubling time (dt, in hours) of a culture of *Staphylococcus haemoliticus* strain 10SBZ1A in the presence of benzo(a)pyrene (BaP), along with various aromatic substrates.** All substrates were used at 40 μmole.l$^{-1}$.

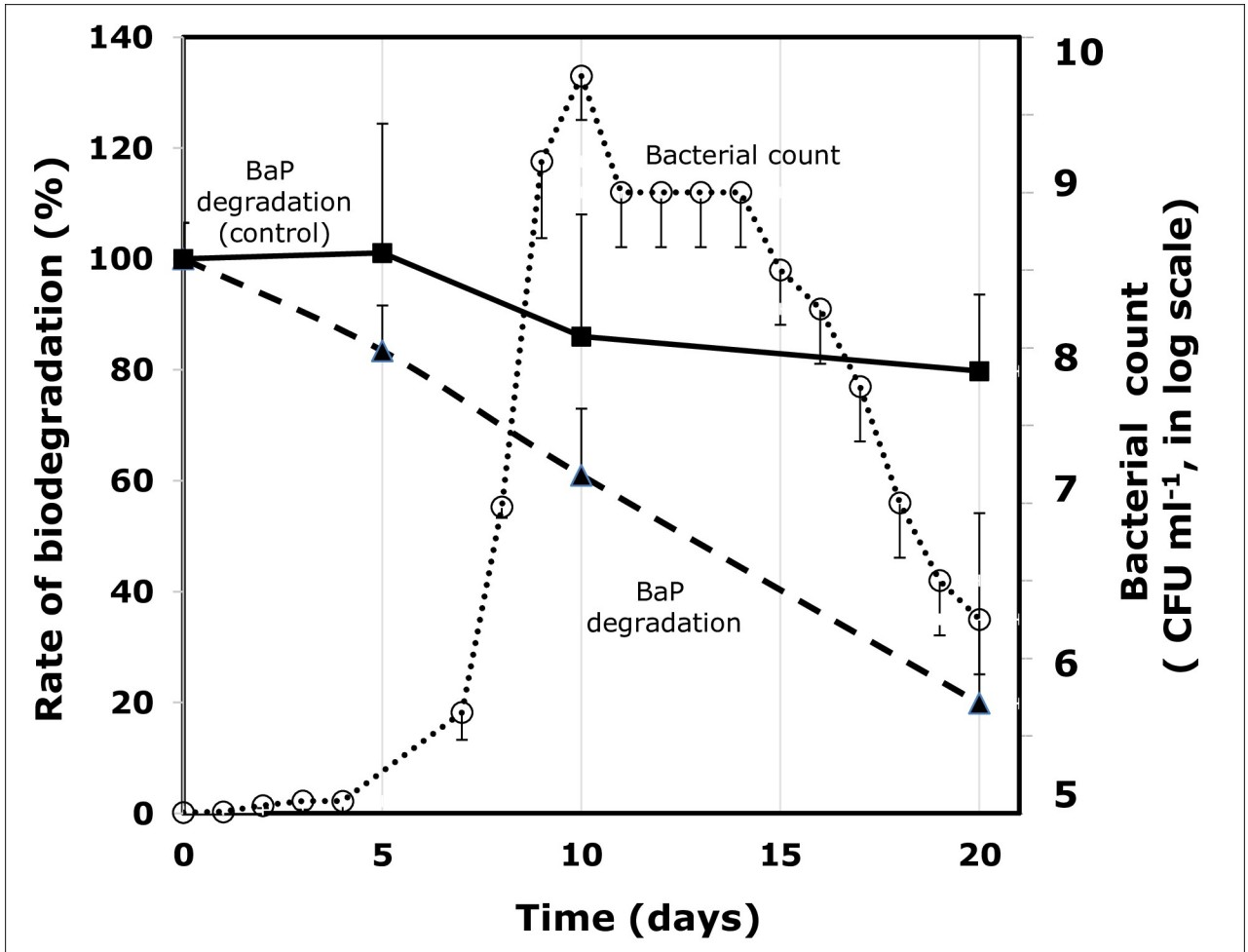

**Fig 4. Quantification of the remaining benzo[a]pyrene (BaP) in a culture of *Staphylococcus haemoliticus* strain 10SBZ1A.** Open circles represent the bacterial growth, and closed squares and closed triangles represent the control and the BaP degradation profiles, respectively.

This degradation rate was more pronounced during the exponential phase of the bacterial growth.

In relation with soil samples spiked with BaP (20 µmol in 100g of soil), the use of this strain 10SBZ1A led to the degradation of 23% of BaP at day 15, and at day 30, 72% of BaP was removed, while in the control samples (soil without bacterial strains), the removal was around 18% only, after day 30.

## BaP Metabolite identification

In an effort to identify the BaP metabolic pathway using the strain 10SBZ1A, we used reversed-phase HPLC to separate metabolites along with TOFMS/MS to analyse BaP metabolites. One of the identified metabolites, at retention time of 22.92 min, had $[M+1]^+$ at *m/z* of 285.0909 with an elemental analysis of $C_{20}H_{12}O_2$, which corresponds to a dihydroxybenzo[a]pyrene (dihydroxy-BaP) [**Table 2**, **S2 Fig**], as reported elsewhere [29]. Dihydrodiol-BaP and BaP-quinone were also detected. A metabolite was observed at a retention time of 24.8 min with $[M+1]^+$ at *m/z* of 317.0811 that corresponds to $C_{20}H_{12}O_4$; it showed a base peak at *m/z* of 299.0717 with elemental analysis of $C_{20}H_{11}O_3$ ($M^{+\cdot}$– 17, loss of OH), and a fragment at *m/z* of

**Table 2. High-resolution mass spectral data for BaP metabolites formed by *Staphylococcus haemoliticus* strain 10SBZ1A.**

| Metabolite | Observed molecular ion mass (calculated) | Retention time (min) | Relative intensity | Molecular formula | Characteristics of major fragments (calculated) |
|---|---|---|---|---|---|
| Dihydroxy-BaP | 285.0909 [M+1] (285.0916) | 22.92 | 41 | $C_{20}H_{12}O_2$ | $C_{20}H_{11}O$ 267.0802 (267.0810) 73, $C_{19}H_{13}O$; 257.0954 (257.0966) 100 |
| BaP-quinone | 283.0756 [M+1] (283.0759) | 26.55 | 93 | $C_{20}H_{10}O_2$ | $C_{19}H_{11}O$ 255.0814 (255.0810) 100; $C_{18}H_{10}$, 226.0783 (226.0782) |
| 4,5-Chrysene-dicarboxylic acid or 4-(8-Hydroxypyren-7-yl)-2-oxobut-3-enoic acid or 4-(7-Hydroxypyren-8-yl)-2-oxobut-3-enoic acid | 317.0811 [M+1] (317.0814) | 24.89 | 9 | $C_{20}H_{12}O_4$ | $C_{20}H_{11}O_3$ 299.0717 (299.0708) 100; $C_{19}H_{11}O_2$ 271.0763 (271.0759) 75. |
| 10-Oxabenzo[*def*]chrysene-9-one or 7-Oxabenzo[def]chrysene-8-one | 269.0607 [M-1] (269.0603) | 24.87 | 71 | $C_{19}H_{10}O_2$ | $C_{19}H_{10}O$, 254.0773 (254.0732) 43 |
| 4-Formylchrysene-5-carboxylic acid | 299.0716 [M-1] (299.0708) | 26.84 | | $C_{20}H_{12}O_3$ | $C_{19}H_{11}O$ 255.0820 (255.0810) 100; $C_{18}H_{11}$ 227.0871 (227.0861) 9 |

271.0763 with elemental analysis of $C_{19}H_{11}O_2$ ($M^{+\cdot}$– 45, loss of $CO_2H$). These fragmentations are consistent with 4,5-chrysene-dicarboxylic acid and 4-(8-hydroxypyren-7-yl)-2-oxobut-3-enoic acid or its isomer 4-(7-hydroxypyren-8-yl)-2-oxobut-3-enoic acid (**Table 2, S2 Fig**).

A metabolite was detected at retention time of 24.87 min with $[M-1]^-$ at *m/z* 269.0607 with an elemental analysis of $C_{19}H_{10}O_2$, which corresponds to either 10-oxabenzo[*def*]chrysene-9-one or its isomer 7-oxabenzo[*def*]chrysene-8-one. This suggests that the detected dihydroxy-BaP could be 9,10-dihydroxy-BaP, which results in a ring opening at C9-C10, or 7,8-dihydroxy-BaP, which results in a ring opening at C7-C8; in both cases, a substituted pyrene is produced. A metabolite was observed at retention time of 26.84 min with $[M-1]^-$ at *m/z* 299.0716 with an elemental analysis of $C_{20}H_{12}O_3$, a base peak at *m/z* of 255.0820 that corresponds to $C_{19}H_{11}O_2$ ($M^{+\cdot}$– 45, loss of $CO_2H$), and a fragment at *m/z* of 227.0861 [$C_{18}H_{11}$, $M^{+\cdot}$– 73, loss of $CO_2H$ and CO]. This fragmentation pattern is consistent with 4-formylchrysene-5-carboxylic acid, which indicates that the observed dihydroxy-BaP could be 4,5-dihydroxy-BaP.

## Discussion

This study has led to the isolation of a *S. heamoliticus*, strain 10BZ1A strain, and bacteria belonging to *Staphylococcus* genera are known to degrade PAHs. For instance, the degradation of naphthalene and phenanthrene have been reported using *Staphylococcus* sp. [30, 31], and that of fluorene using *Staphylococcus auricularis* [32]. Likewise, a strain of *Staphylococcus aureus* was shown to grow in the presence of crude oil [33]. The current data show the ability of a bacterial species of this genus to degrade the HMW-PAH BaP. This study also showed that the increase in BaP concentration is associated with a decrease in 10BZ1A growth, which is consistent with previously reported studies. This includes anthracene using *Bacillus licheniformis* [34], and a co-culture of *Ralstonia pickettii* and *Thermomonas haemolytica* [28]; phenanthrene on a co-culture of *Pseudomonas citronellolis* and *S. maltophilia* [28]; anthracene and pyrene on *Ochrobactrum* sp. [25]; pyrene on *Achromobacter xylosoxidans*, and on the halophilic strains of *Halomonas shengliensis* and *Halomonas smyrnensis* [27, 35].

The results of this work also indicate that *S. haemoliticus* strain has an optimum pH of 7 in degrading BaP. Several reports indicate that the optimum pH range for the degradation of PAHs fall between 6 and 8, and the neutral pH being the most common [23, 36]. Nevertheless, efficient degradation of PAHs in acidic and alkaline conditions have also been reported [23]. So far, reports on active microorganisms degrading BaP has been centered at pH 7. In relation with temperature, several species of thermophilic bacteria have been reported to degrade PAHs at temperatures between 50 and 70 $^{\circ}C$ [37], however, limited work has been reported on BaP. A consortium of *Geobacillus* spp. and *Thermus* sp. could degrade BaP at 60–70 $^{\circ}C$, but

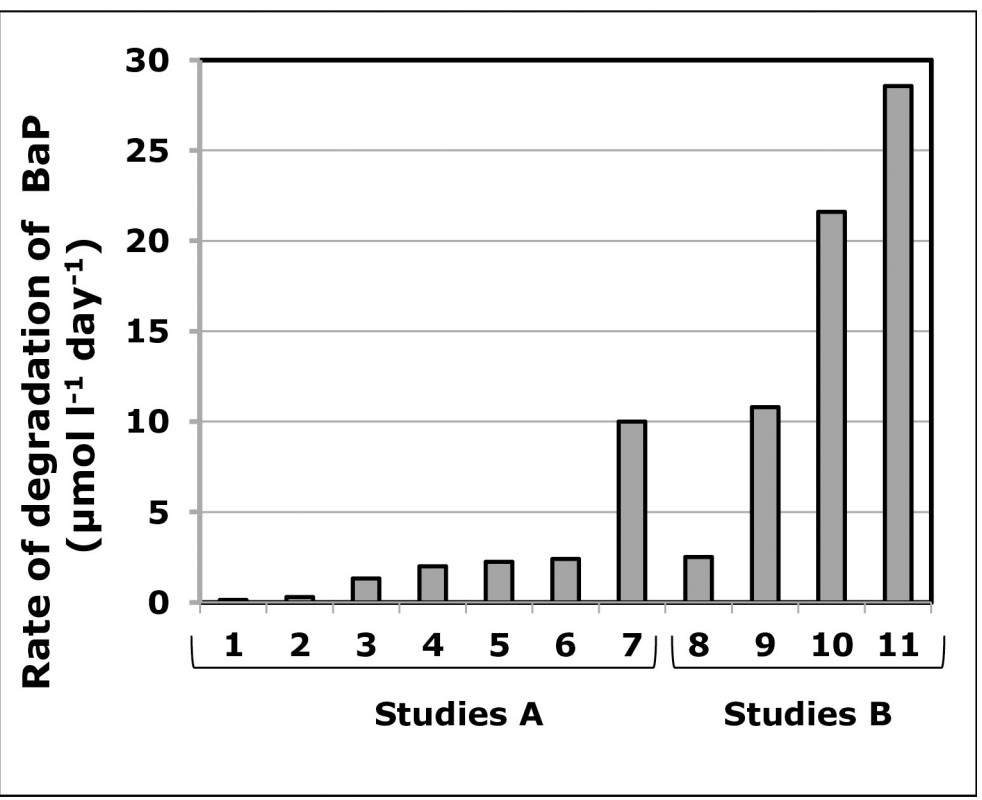

**Fig 5. Values of degradation rates of benzo[a]pyrene (BaP) reported in various studies. Studies A** were cultures of single strains, in minimum mineral media (MM); **studies B** consisted of consortia of bacteria in MM or rich medium, or single bacterial strain but in rich media. References are listed in the text, in the Results/Discussions section.

only if hexane was added as a growth substrate, a classical approach of cometabolism [6, 38]. So far, only one bacterial strain, the thermophilic *Bacillus licheniformis* M2-7, has been reported to degrade BaP as a sole substrate at 50 °C [22]. In the current work, the investigation on the temperature effect has shown the optimum range of 35–40 °C.

Bacteria of *Staphyloccocus* genus are known to be halotolerant, with an optimum range of 7.5–10% NaCl [39, 40]. Moreover, the strain 10SBZ1A grows at a similar salinity level, when cultured in the presence of BaP as a sole source of carbon. As stated earlier, several species of halophilic bacteria and archaea can degrade petroleum products, including PAHs such as naphthalene, phenanthrene, fluorene and pyrene [3, 23, 24, 41]. The degradation of BaP in saline conditions has been reported in the context of cometabolism (in the presence of phenanthrene), with the use of a consortium of bacteria *(Achromobacter* sp. AYS3, *Marinobacter* sp. AYS4 and *Rhodanobacter* sp. AYS5), in a medium containing 3–9% NaCl [26]. Recently, another consortium consisting of *Ochrobactrum anthropi*, *Stenotrophomonas acidaminiphila*, and *Aeromonas salmonicida* was shown to degrade BaP in seawater, which has a salinity level of about 3–5% NaCl [42]. So far, the use of a single strain for BaP degradation in halophilc conditions has only been reported once, using *Ochrobactrum* sp. VA1 strain [25]. However, careful analysis of these data showed that the tested NaCl concentrations were relatively low, 3% NaCl only [25]. The current work reports the isolation and characterisation of a single bacterial strain that can actively degrade BaP at high salinity of 10% NaCl.

In general, bacteria degrade PAHs in a stepwise ring-opening process, and the last aromatic intermediate being a mono-aromatic compound. As discussed earlier, the less complex the

PAH is, the easier the degradation is. Thus, a bacterium that degrades a given PAH can also utilise a PAH of lower molecular weight [13]. The reported results confirmed that the lower the molecular weight of PAHs, the faster the degradation, and the aliphatic ones are more degradable that the aromatic ones. Thus, this strain could be used for the removal of both aliphatic and aromatic pollutants. The results of this investigation are also in agreement with reports indicating the ability of BaP-degrading bacteria to utilise PAHs of lower molecular weights, as it has been shown in *Ochrobactrum* sp. BAP5 [43], *Ochrobactrum* sp. VA1 [25], *Hydrogenophaga* PYR1 [44], *Cellulosimicrobium cellulans* CWS2 [45], *Rhizobium tropici* CIAT 899 [46], *Klebsiella pneumonia PL1* [47] and *Pseudomonas* sp. JP1 [48].

The rate of BaP degradation using 10SBZ1A was compared to those reported in similar studies (which were all carried out using non-halophilic microbes), and these results are summarised in **Fig 5**. Of particular importance are studies 1–7 (**Fig 5**) that involved single bacterial strains, cultured in minimum mineral medium containing BaP as the sole source of carbon, as in the current study. In these studies, BaP degradation rates fell between 0.04–0.3 $\mu$mol l$^{-1}$ day$^{-1}$, and the rate reported in the current work falls within this range [17, 25, 43, 45, 47–49]. Higher rates were reported, however, they were associated with the use of either a consortium of bacterial strains or rich culture medium (Studies 8–11, **Fig 5**) [21, 42, 46, 50]. Thus, in similar experimental conditions, the halophilic strain 10SBZ1A had a BaP degradation rate in the range of those of non-halophilic bacteria.

Liquid chromatography-tandem mass spectrometry analysis led to the identification of several metabolites, including dihydroxy-BaP and dihydrodiol-BaP. In aerobic condition, the first and the most important step in the degradation of aromatic compounds is the generation of dihydroxy-aromatic intermediates from the action of dioxygenase enzymes [11–13]. For instance, this enzyme, owing to its central role in PAH degradation, has been used as genetic probe to track and identify pyrene-degrading bacteria in various environments [51, 52]. The metabolite, dihydroxy-BaP, reported in the current study, has also been identified in two bacterial strains, *Beijerinckia* B-836 as 9,10-dihydroxy-dihydroBaP [53], and *Mycobacterium* RJGII-135s as 7,8-dihydroxyBaP [54]. Following BaP dihydroxylation, ring opening will occur, generating derivatives of pyrene if dihydroxylation occurs at positions C7-C8 or C9-C10, or chrysene derivatives if it occurs at positions C4-C5. In the current study, the exact position of the dihydroxylation could not be resolved because of lack of reference metabolites, however, the identification of 4,5-chrysene-dicarboxylic acid, and that of 4-formylchrysene-5-carboxylic acid indicates that the dihydroxylation of BaP occurred at C4-C5. Chrysene analog 4,5-chrysene-dicarboxylic acid has been previously reported in *Mycobacterium* RJGII-135s [54]. Interestingly, in addition to chrysene derivatives, the pyrene intermediate 4-(8-hydroxypyren-7-yl)-2-oxobut-3-enoic acid (or its isomer 4-(7-hydroxypyren-8-yl)-2-oxobut-3-enoic acid) is a possible metabolite for BaP using the strain 10SBZ1A, indicating the ring opening option at C7-C8, thus BaP dihydroxylation of C7-C8 is not eliminated. It is worth mentioning that these two isomers and 4,5-chrysene-dicarboxylic acid have the same exact mass and have been reported to exhibit similar mass fragmentation pattern [54], which hamper their distinction from each other specially with lack of standards. BaP dihydroxylation can occur at more than one position, a feature that has already been reported [54, 55]. Based on the above analysis, we propose the BaP degradation pathways shown in **Fig 6**. Future investigations of metabolites that include the synthesis of specific metabolites is crucial to unravel more details about the metabolic pathway of BaP by the strain 10SBZ1A.

## Conclusion

Several investigations have been dedicated on the biodegradation of HMW-PAHs, including BaP, however most of that work has, so far, focused on the mesophilic bacteria. The current

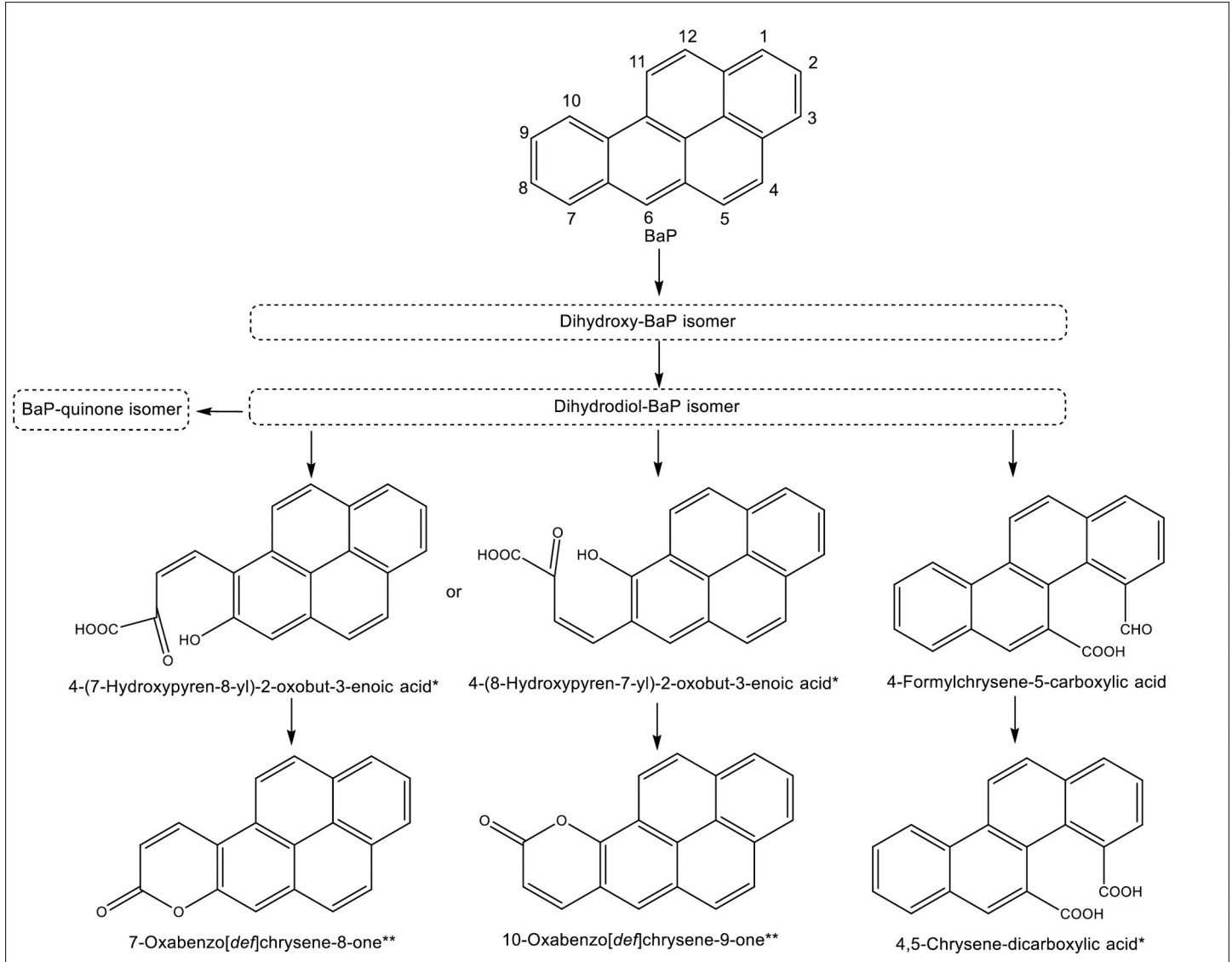

**Fig 6. Propose pathways for the degradation of BaP by *Staphylococcus haemoliticus* strain 10SBZ1A.** *Compounds with identical exact molar mass. ** Compounds with identical exact molar mass.

work reports, for the first, the isolation and characterisation of a bacterial strain that is capable to degrade BaP at saline concentration as high as 10% NaCl, in mineral culture medium and in soil samples. The strain can also degrade PAHs of lower molecular weight, along with aliphatic compounds, and was active at 37 °C and neutral pH. In addition, the report shows that this halophilic bacterium metabolises BaP through the typical aromatic-ring-hydroxylation dioxygenation, followed by ring opening, as it has been shown in many other PAH degrading bacteria. As discussed earlier, the degradation of PAH has been reported in several bacterial strains belonging to *Staphylococcus* genus, including the species *S. aureus* and S. *auriculans*, and in the current study, *S. heamoliticus*. Although these strains cannot be used in bioremediation of contaminated environments because of their medical importance, however they can be of use in deciphering the mechanisms of PAH degradation.

## Supporting information

**S1 Fig. *Staphylococcus haemoliticus* strain 10SBZ1A colony forms. (A)**, from light microscope and [**B**], from scanning electron microscopy (SEM).
(TIF)

**S2 Fig. Spectra of several Benzo(a)pyrene metabolites from *Staphylococcus haemoliticus* strain 10SBZ1A, using liquid chromatography-tandem mass spectrometry (LC-Mass Spec).**
(ZIP)

**S1 Data.**
(XLSX)

## Author Contributions

**Conceptualization:** Alexis Nzila, Musa M. Musa.

**Data curation:** Musa M. Musa, Qing X. Li.

**Formal analysis:** Alexis Nzila, Musa M. Musa, Marwan Al-Momani, Qing X. Li.

**Funding acquisition:** Musa M. Musa.

**Investigation:** Alexis Nzila, Saravanan Sankara, Lei Xiang.

**Methodology:** Alexis Nzila, Musa M. Musa, Lei Xiang, Qing X. Li.

**Project administration:** Alexis Nzila, Musa M. Musa.

**Resources:** Musa M. Musa, Saravanan Sankara, Qing X. Li.

**Software:** Marwan Al-Momani, Lei Xiang.

**Validation:** Alexis Nzila, Musa M. Musa, Qing X. Li.

**Writing – original draft:** Alexis Nzila, Musa M. Musa.

**Writing – review & editing:** Alexis Nzila, Musa M. Musa, Marwan Al-Momani, Lei Xiang, Qing X. Li.

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
