## [Decision Letter · Decision Letter 0]

20 Jan 2021

PONE-D-20-38549

Degradation of benzo[a]pyrene by the halophilic bacterium strain Staphylococcus haemoliticus, 10SBZ1A

PLOS ONE

Dear Dr. Nzila,

Thank you for submitting your manuscript to PLOS ONE. After careful consideration, we feel that it has merit but does not fully meet PLOS ONE’s publication criteria as it currently stands. Therefore, we invite you to submit a revised version of the manuscript that addresses the points raised during the review process.

We look forward to receiving your revised manuscript.

Kind regards,

Pankaj Kumar Arora

Academic Editor

PLOS ONE

Journal Requirements:

3. Please include a copy of Table 1 which you refer to in your text on page 10.

Reviewers' comments:

Reviewer's Responses to Questions

**Comments to the Author**

1. Is the manuscript technically sound, and do the data support the conclusions?

Reviewer #1: Yes

Reviewer #2: Yes

2. Has the statistical analysis been performed appropriately and rigorously? 

Reviewer #1: Yes

Reviewer #2: Yes

3. Have the authors made all data underlying the findings in their manuscript fully available?

Reviewer #1: Yes

Reviewer #2: No

4. Is the manuscript presented in an intelligible fashion and written in standard English?

Reviewer #1: Yes

Reviewer #2: Yes

5. Review Comments to the Author

Reviewer #1: A halophilic BaP degradation strain was isolated, and its characteristics and a metabolic pathway of BaP was investigated and proposed. As the first report of a halophilic BaP degrading bacterial strain at salinity > 5% NaCl, this paper has some innovation. However, some details are still need to be addressed. Please follow the comments:

1. Abstract Line 42-43, Please carefully check the results of doubling time ‘2-10h’ and degradation rate ‘0.2 μmol/l’. It is inconsistent with the text.

2. Abstract Line 42-44, “The strain 10SBZ1A degraded BaP at a rate of 0.2 μmol/l per day” “The optimum conditions of growth were 37 ºC, 10% NaCl (w/v) and pH 7.” It’s better to change the order of the two sentences. It should be the degradation rate under optimum conditions.

3. Materials and methods Line 202-208, What’s the salinity of the soil sample? Please add the operation conditions of this experiment, including temperature, salinity, and pH.

4. Introduction Line 60-66, this part introduced the background of this study. I advise the authors to supply some references of benzo[a]pyrene as the main PAHs components existed in the produced water. In addition, some references are also needed to support the high salinity of PW, even up to 30% NaCl.

Reviewer #2: Comments:

1. Delete the word "the" from the title and there should be "bacterial" in place of "bacterium".

2. The end of the sentence is incomplete "and among them, Benzo[a]pyrene (BaP)" at Line No. 34-35. Rewrite.

3. Cite the following reference at the end of the sentenece "The biodegradation of HMW-PAHs is challenging........difficult it is to degrade": Arora (2020), doi: 10.3389/fbioe.2020.570307

4. Delete "also" from the sentence at Line no. 35.

5. Rephrase the sentence " that can reach up to 30% NaCl" at Lie No. 35-36.

6. There should be "organic contaminants" in place of "PAHs" at Line no. 75.

7. Mention the purity status of the followings: pyrene, anthracene, phenanthrene, naphthalene, phthalic

120 acid, salicylic acid, catechol; palmitic, oleic and stearic acids.

8. Write the model, make and country of origin for GC at Line No. 208.

9. Why the chromatograms of liquid chromatography-high resolution tandem mass spectrometry, gas chromatography-mass spectrometry and HPLC is not included in the main text. Include all these chromatogram and supporting information in the main text of the manuscript.

6. PLOS authors have the option to publish the peer review history of their article (what does this mean?). If published, this will include your full peer review and any attached files.

Reviewer #1: No

Reviewer #2: No

---

## [Author Response · Author response to Decision Letter 0]

4 Feb 2021

A file has been uploaded.. in addition, below the same information

PONE-D-20-38549

Degradation of benzo[a]pyrene by the halophilic bacterium strain Staphylococcus haemoliticus, 10SBZ1A

PLOS ONE

POINT BY POINT RESPONSE TO THE REVIEWERS’ COMMENTS

OUR RESPONSE: Our response: we have rechecked our manuscript about the style requirement

 OUR RESPONSE: We have uploaded an excel file containing the necessary set of data to replicate our findings. 

2. Please include a copy of Table 1 which you refer to in your text on page 10.

OUR RESPONSE: This has been an oversight, the Table has been added. 

4. Have the authors made all data underlying the findings in their manuscript fully available?

OUR RESPONSE: As mentioned earlier, we have uploaded an excel file containing the necessary set of data to replicate our findings.

Reviewer 1 

1. Abstract Line 42-43, Please carefully check the results of doubling time ‘2-10h’ and degradation rate ‘0.2 μmol/l’. It is inconsistent with the text.

OUR RESPONSE: We thank the reviewer for this remark. Yes, there was inconsistency with the text and the figures. This has been changed accordingly. 

2. Abstract Line 42-44, “The strain 10SBZ1A degraded BaP at a rate of 0.2 μmol/l per day” “The optimum conditions of growth were 37 ºC, 10% NaCl (w/v) and pH 7.” It’s better to change the order of the two sentences. It should be the degradation rate under optimum conditions.

OUR RESPONSE: We agree with the suggestion. Therefore we have changed the order of these 2 sentences. 

3. Materials and methods Line 202-208, What’s the salinity of the soil sample? Please add the operation conditions of this experiment, including temperature, salinity, and pH.

Our response: The soil was not used dried. It was kept in the medium solution (BH), pH 7, salinity 5%. We have clarified this in the text. 

4. Introduction Line 60-66, this part introduced the background of this study. I advise the authors to supply some references of benzo[a]pyrene as the main PAHs components existed in the produced water. In addition, some references are also needed to support the high salinity of PW, even up to 30% NaCl.

OUR RESPONSE: We quoted reference (Ref.1), that is relevant in relation with produced water (PW). In light of the reviewer comment, we have added an additional reference (Ref.2), and this reference provides information that PAHs are part of important pollutants found in PW. 

Reviewer #2: 

1. Delete the word "the" from the title and there should be "bacterial" in place of "bacterium".

OUR RESPONSE: Done

2. The end of the sentence is incomplete "and among them, Benzo[a]pyrene (BaP)" at Line No. 34-35. Rewrite.

OUR RESPONSE: We have removed “among them” and replace it with “including”. The sentence is not clearer.

3. Cite the following reference at the end of the sentence "The biodegradation of HMW-PAHs is challenging........difficult it is to degrade": Arora (2020), doi: 10.3389/fbioe.2020.570307

OUR RESPONSE: Done

4. Delete "also" from the sentence at Line no. 35.

OUR RESPONSE: Done

5. Rephrase the sentence " that can reach up to 30% NaCl" at Lie No. 35-36.

OUR RESPONSE: This has been done, by replacing" that can reach up to 30% NaCl" by “which can be as high as 30% NaCl”

6. There should be "organic contaminants" in place of "PAHs" at Line no. 75.

OUR RESPONSE: In this sentence, we are referring to a specific type of the organic pollutants, which are PAHs. Thus, replacing PAHs will make the sentence not clear.. So we suggest to keep it. 

7. Mention the purity status of the followings: pyrene, anthracene, phenanthrene, naphthalene, phthalic

120 acid, salicylic acid, catechol; palmitic, oleic and stearic acids.

OUR RESPONSE: All chemicals were of HPLC, analytical grades. We have added this information in the text. 

8. Write the model, make and country of origin for GC at Line No. 208. 

OUR RESPONSE: We have provided the necessary additional information of the GC equipment we have used

9. Why the chromatograms of liquid chromatography-high resolution tandem mass spectrometry, gas chromatography-mass spectrometry and HPLC is not included in the main text. Include all these chromatogram and supporting information in the main text of the manuscript.

Our response: This information has been added as Supplementary material. Adding this information in the text is going to make it unnecessary long. Such information is appropriate to be added as Supplementary material.. Thus, for the conciseness of the manuscript, we suggest to keep it as Supplementary material. 

END DOCUMENT

---

## [Editor Report · Decision Letter 1]

8 Feb 2021

PONE-D-20-38549R1

Degradation of benzo[a]pyrene by halophilic bacterial strain Staphylococcus haemoliticus, 10SBZ1A

PLOS ONE

Dear Dr. Nzila,

Thank you for submitting your manuscript to PLOS ONE. After careful consideration, we feel that it has merit but does not fully meet PLOS ONE’s publication criteria as it currently stands. Therefore, we invite you to submit a revised version of the manuscript that addresses the points raised during the review process.

We look forward to receiving your revised manuscript.

Kind regards,

Pankaj Kumar Arora

Academic Editor

PLOS ONE

Additional Editor Comments (if provided):

Whole manuscript should be edited for grammatical and English languages. Some examples are:

1. There are two full-stop line number 44.

2. Line 33, produced water is not scientific term. Use petroleum waste effluent or something else in whole manuscript.

3. use word " Strain before 10SBZ1A in whole manuscript.

4. Line 258- Bacterial DNA was isolated and purified.

5. Check other errors.

---

## [Author Response · Author response to Decision Letter 1]

10 Feb 2021

Point by point response to the editor comments

1. There are two full-stop line number 44.

Our response: This has been removed

2. Line 33, produced water is not scientific term. Use petroleum waste effluent or something else in whole manuscript.

Our response: The term produced water is used in many publications, for instance, in Ref1 &2, the term “produced water” is even mentioned in the titles.. However, in light of the editor comments, we have mentioned that produced water is same as petroleum waste effluent (line 34 and 61), that we have replaced “produced water” by “petroleum waste effluent (PWE)” throughout the text.

3. use word " Strain before 10SBZ1A in whole manuscript.

Our response: This has been changed throughout the text, the tables and the figures

4. Line 258- Bacterial DNA was isolated and purified.

Our response: This has been done

5. Check other errors.

Our response: We have cross-checked the all text for mistakes.

END DOCUMENT

---

## [Editor Report · Decision Letter 2]

12 Feb 2021

Degradation of benzo[a]pyrene by halophilic bacterial strain Staphylococcus haemoliticus strain 10SBZ1A

PONE-D-20-38549R2

Dear Dr. Nzila,

We’re pleased to inform you that your manuscript has been judged scientifically suitable for publication and will be formally accepted for publication once it meets all outstanding technical requirements.

Kind regards,

Pankaj Kumar Arora

Academic Editor

PLOS ONE
---

## [Editor Report · Acceptance letter]

17 Feb 2021

PONE-D-20-38549R2 

Degradation of benzo[a]pyrene by halophilic bacterial strain *Staphylococcus haemoliticus strain 10SBZ1A *

Dear Dr. Nzila:

I'm pleased to inform you that your manuscript has been deemed suitable for publication in PLOS ONE. Congratulations! Your manuscript is now with our production department. 

Kind regards, 

on behalf of

Dr. Pankaj Kumar Arora 

Academic Editor

PLOS ONE